# Incidence trends of gastric cancer in the United States over 2000–2020: A population-based analysis

**Armin Aslani[1], Amirali Soheili[2], Seyed Ehsan Mousavi[1,3]\*, Ali Ebrahimi[2], Ryan Michael Antar[4], Zahra Yekta[5], Seyed Aria Nejadghaderi[6,7]\***

**1** Social Determinants of Health Research Center, Department of Community Medicine, Faculty of Medicine, Tabriz University of Medical Sciences, Tabriz, Iran, **2** Medical Student Research Committee, School of Medicine, Shahid Beheshti University of Medical Sciences, Tehran, Iran, **3** Neurosciences Research Center, Aging Research Institute, Tabriz University of Medical Sciences, Tabriz, Iran, **4** The George Washington University School of Medicine & Health Sciences, Washington, DC, United States of America, **5** Calaveras County Department of Health, Calaveras County, California, United States of America, **6** HIV/STI Surveillance Research Center, and WHO Collaborating Center for HIV Surveillance, Institute for Futures Studies in Health, Kerman University of Medical Sciences, Kerman, Iran, **7** Systematic Review and Meta-analysis Expert Group (SRMEG), Universal Scientific Education and Research Network (USERN), Tehran, Iran

\* ariang20@gmail.com (SAN); drehsanmousavii@gmail.com (SEM)

**Data Availability Statement:** All data are within the article and its supporting information files. The data used in this study are available from the

## Abstract

### Background

Gastric cancer ranks among the top cancers in terms of both occurrence and death rates in the United States (US). Our objective was to provide the incidence trends of gastric cancer in the US from 2000 to 2020 by age, sex, histology, and race/ethnicity, and to evaluate the effects of the COVID-19 pandemic.

### Methods

We obtained data from the Surveillance, Epidemiology, and End Results 22 program. The morphologies of gastric cancer were classified as adenocarcinoma, gastrointestinal stromal tumor, signet ring cell carcinoma, and carcinoid tumor. We used average annual percent change (AAPC) and compared pairs using parallelism and coincidence. The numbers were displayed as both counts and age-standardized incidence rates (ASIRs) per 100000 individuals, along with their corresponding 95% confidence intervals (CIs).

### Results

Over 2000–2019, most gastric cancers were among those aged ≥55 years (81.82%), men (60.37%), and Non-Hispanic Whites (62.60%). By histology, adenocarcinoma had the highest incident cases. During the COVID-19 pandemic, there was a remarkable decline in ASIRs of gastric cancer in both sexes and all races (AAPC: -8.92; 95% CI: -11.18 to -6.67). The overall incidence trends of gastric cancer were not parallel, nor identical.

Surveillance, Epidemiology, and End Results Program (SEER) database at https://seer.cancer.gov/data/.

**Funding:** The author(s) received no specific funding for this work.

**Competing interests:** The authors have declared that no competing interests exist.

## Conclusions

The incidence of gastric cancer shows notable variations by age, race, and sex, with a rising trend across ethnicities. While the overall incidence has declined, a noteworthy increase has been observed among younger adults, particularly young Hispanic women; however, rates decreased significantly in 2020.

## Introduction

Gastric cancer poses a considerable health issue worldwide and stands as a primary contributor to cancer-related mortalities [1]. It is the fifth cancer in incidence and the fourth leading cause of cancer mortality among both sexes worldwide [2]. In 2020, in the United States (US), 22425 new cases of gastric cancer were reported, causing more than 11000 deaths [3]. Despite recent advancements in diagnostic and therapeutic modalities, the survival rates for gastric cancer remain suboptimal, with a five-year survival rate of 36.4% in the US [4]. The epidemiology of gastric cancer is characterized by marked geographical variations, with the highest incidence rates observed in Eastern Asia and Eastern Europe. In contrast, incidence rates in Northern America and Europe are typically minimal and comparable to those observed in African areas [2]. Gastric cancer is approximately twice as common in men as in women and typically affects individuals over the age of 60 years, both globally and in the US [2, 3, 5]. The classification of gastric cancer is based on the location, histology, and tumor stage [6]. Histologically, gastric cancers are primarily classified into adenocarcinoma, which accounts for approximately 90% of cases. Adenocarcinoma itself has several subtypes, including tubular, papillary, mucinous, and poorly cohesive types. In addition to adenocarcinoma, there are other less common histologic variants such as gastrointestinal stromal tumors (GISTs), lymphomas, and carcinoid tumors [6, 7].

The unprecedented COVID-19 pandemic has significantly affected healthcare systems globally, disrupting the diagnosis, treatment, and screening of a myriad of diseases, including gastric cancer [8]. There is currently limited research on the effects of COVID-19 on gastric cancer incidence [9]. In addition, the pandemic has led to delayed diagnosis and treatment, suspension or disruption of gastric cancer screening programs, and changes in incidence reporting programs [10, 11]. The COVID-19 pandemic induced lifestyle alterations, which could lead to increased gastric cancer risk [8].

Remarkably, while the incidence of gastric cancer has been declining globally over the past few decades, it remains a significant health burden [1, 12]. In the US, the age-adjusted incidence rates of gastric cancer show the same pattern [13]. Recent studies suggest gastric cancer is increasing in certain US populations like non-Hispanic Whites (NHWs) aged <50 years [14, 15]. Nevertheless, these studies are indiscriminate about different histological types of gastric cancer. Additionally, the incidence trends of gastric cancer in the US have been reported previously. However, they need to be updated [16, 17]. Moreover, past research, including studies conducted prior to 2020, did not access to the data after the COVID-19 pandemic and has not assessed the impact of the COVID-19 pandemic on the incidence rates of gastric cancer [16, 17]. Therefore, reporting the most up-to-date incidence trends of gastric cancer is crucial for understanding its etiology, identifying risk factors, and planning effective cancer control strategies. The Surveillance, Epidemiology, and End Results (SEER) program provides information that can be used to analyze cancer incidence trends over time [18]. Herein, we aimed to examine the incidence trends of gastric cancer in the US from 2000 to 2020, categorizing by age,

sex, histology, and race/ethnicity, using data from the SEER program. Furthermore, we evaluated the influence that COVID-19 has had on the incidence trends. By examining these trends, the study provided preliminary evidence of how the pandemic may have disrupted healthcare services and diagnostic processes, potentially affecting reported incidence rates in 2020. This analysis contributes to the existing literature by enhancing the understanding of how demographic and histological factors influence gastric cancer trends and by offering insights into the pandemic's impact on cancer reporting. Furthermore, the findings have implications for public health policies and cancer control strategies, particularly in addressing healthcare disparities and improving access to timely diagnosis and treatment. Overall, this study sets the stage for future research to explore the long-term effects of the COVID-19 pandemic on cancer trends and to better understand the factors influencing gastric cancer incidence.

## Methods

### Data source

The SEER program, established by the National Cancer Institute, is an extensive population-based database concerning cancer within the US. The SEER 22 registries encompass nearly 48% of the US population, providing survival outcomes and stage of cancer at initial diagnosis [19]. The SEER program collects data pertaining to the patient's demographic characteristics, initial tumor location, tumor morphology, stage of diagnosis, initial treatment received, and subsequent vital status monitoring [19]. In April of 2023, the SEER 22 database became available, containing data reported in November 2022. This was employed in this study to estimate the incidence rates and annual percent changes (APCs) of gastric cancer within the period of 2000–2020 [20, 21]. The SEER 22 database was utilized in accordance with the SEER Research Data Agreement covering data from 1975 to 2020 (submitted in November 2022) [22], and cancer statistics were published following the SEER 22 guideline [23]. Using this database allowed us to perform a thorough examination of gastric cancer trends over two decades, providing valuable insights into incidence patterns and variations.

Given the nature of this study and the data which is available at http://seer.cancer.gov/data/ , there was no requirement for institutional review board submission and do not require patient informed consent. Access to the SEER data was in accordance with the SEER data agreement. All data from SEER database were fully anonymized.

### Variables and cancer definitions

Cancer cases in this study were provided by raw counts and percentages, whereas the incidence rate is reported as cases per 100000 people. The APCs of gastric cancer throughout a specific time revealed variance at a constant proportion of the rate observed in the previous year. The average annual percent changes (AAPCs) represent the mean of multiple APCs during a specific period. The participants were divided into groups based on ethnicity: NHW, Non-Hispanic Black (NHB), and Hispanic. However, because of the restricted number of instances, the racial and ethnic categories of American Indian/Alaska Native, Native Hawaiian, and Asian/Pacific Islander were solely employed for determining parameters relevant to all combined races and ethnicities. Gastric cancer patients were identified using the International Classification of Diseases for Oncology version 3. The morphologies of gastric cancer were classified as adenocarcinoma (codes 8140 to 8147, 8210, 8211, 8214, 8220, 8221, 8255, 8260 to 8263, 8310, 8480, 8481, 8570, and 8574 to 8576), GIST (code 8936), signet ring cell carcinoma (code 8490), and carcinoid tumor (codes 8240 to 8246). The study included all primary tumors, not just the first primary tumors, to provide a comprehensive assessment of gastric cancer incidence. This approach captures the full scope of the disease burden within the

population and ensures that the analysis reflects the total number of gastric cancer cases as reported in the SEER dataset.

## Statistical analysis

The study utilized the Research Limited-Field data of the SEER 22 dataset, incorporating a database with delayed-adjustments covering the period from 2000 to 2020 [20], which was obtained from SEER*Stat, version 8.4.1.2 [24], to calculate the delayed age-standardized incidence rate (ASIR) of gastric cancer. The Research Limited-Field data accounts for reporting delays and adjustments, which helps in refining the estimates of cancer incidence by incorporating anticipated future revisions to the data, such as updates and corrections [25]. These modified counts and the related delayed model can be used to detect current cancer trends more precisely [25]. The selection of cases was restricted to individuals diagnosed with malignant cancer and whose age at diagnosis was known. Subsequently, the delayed adjustment was applied, incorporating adjustment factors for variables such as cancer site, registry, age group, race and ethnicity, and year of diagnosis [26, 27]. Furthermore, the SEER 22 Research Limited-Field data from 2000 to 2020 was used to determine the ASIRs of gastric cancer subtypes [21], retrieved from the SEER*Stat, version 8.4.1.2 [24]. The Tiwari technique was used to estimate the ASIRs based on the 2000 US standard population and the accompanying 95% confidence intervals (CIs) [28] using the SEER*Stat version 8.4.1.2 [24]. The Tiwari technique is particularly useful in producing stable estimates by smoothing the data, thus reducing the variability caused by small numbers and fluctuations in annual data

To estimate the APCs, AAPCs [29], joinpoint regression modeling, the parallelism test, and the coincident test [30] for ASIRs [31], the Joinpoint Regression Program, version 5.0.2, was used [32]. The program's weighted Bayesian Information Criteria approach was applied to select the optimal number of joinpoints, balancing model fit with complexity to avoid overfitting. Joinpoint regression modeling was utilized to detect significant changes in trends over time. This technique identifies points in time where the trend shifts significantly, allowing for the detection of changes in the trajectory of cancer incidence. The onset of the COVID-19 pandemic occurred in 2020. To calculate the APCs of gastric cancer ASIRs, least-squares regression lines were fitted on the natural logarithm of ASIRs, using the diagnosis year as the predictor variable. A minimum of two observations was required between two joinpoints, and there needed to be a minimum of two observations from a joinpoint to either end of the dataset. The weighted Bayesian Information Criteria approach was used for selecting models [33]. The empirical quantile method was utilized to calculate the 95% CI of AAPCs, providing a range that reflects the precision of the average trend estimates over the study period [34]. The parallelism test was utilized through pairwise comparison to assess whether the trends of the two groups exhibited similarity over time [30]. Additionally, a pairwise comparison utilizing the coincidence test was conducted to ascertain if the rates of the two groups remained consistent over time. The use of advanced statistical techniques such as delayed-adjustment models, joinpoint regression, and empirical quantile methods allows for a detailed analysis of gastric cancer incidence trends, considering data reporting delays, trend changes, and inter-group comparisons.

## Results

### Gastric cancer

**Overall incidence.** Overall, 227339 cases of gastric cancer were reported in the US among all ages between 2000 and 2019. Adenocarcinoma (68.98%) represented the most common subtype in our analysis (Fig 1). The most common cases in our study included those aged ≥55

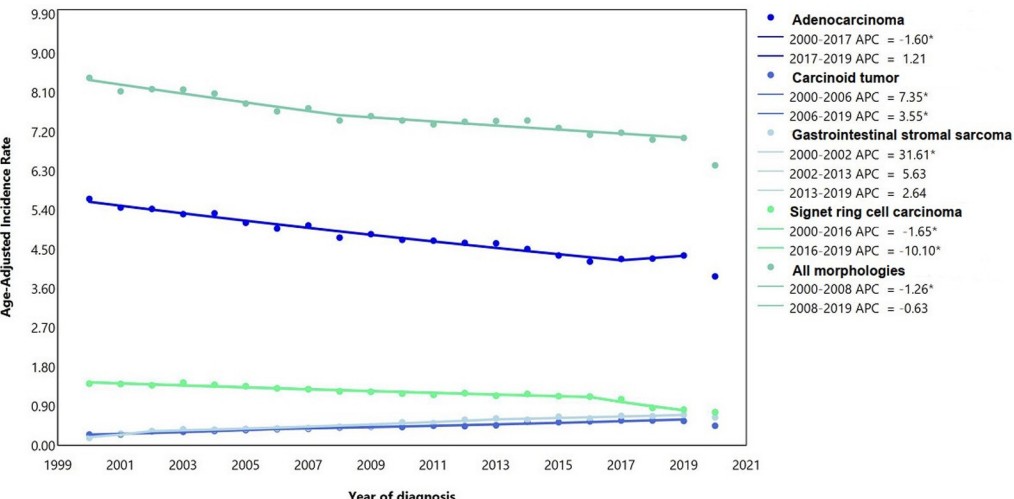

**Fig 1. Age-adjusted incidence rate of gastric cancer over 2000–2019 and in 2020 in the United States, by histologic type.** APC: annual percent change. * Represent p-value less than 0.05.

years (81.82%), men (60.37%), and NHWs (62.60%) (Table 1 and S1–S3 Figs). Between 2000 and 2019, the delayed ASIRs of gastric cancer per 100000 for men and women were 10.30 (95% CI: 10.25 to 10.36) and 5.40 (5.37 to 5.44), respectively, which showed -1.32% (-1.47 to -1.16) and -0.42% (-0.58 to -0.25) decreases in men and women over the 2000–2019 study period, respectively (Table 1 and S1 Fig). By race/ethnicity, NHBs (15.83; [15.59 to 16.08]) and Hispanics (14.13; [13.93 to 14.32]) had the highest delayed ASIRs among men (Table 1 and S2 Fig). S1 and S2 Tables present the overall parallel and identical trends of gastric cancer and its subtypes, respectively.

**Older adults (Ages≥55).** Throughout the 2000 to 2019 study period, there were an overall 186012 gastric cancer cases in those aged≥55 years. Adenocarcinoma was the most common subtype (72.38%) in this age group. Also, most patients were males (60.94%) and NHWs (65.82%) (not shown in the table). Delayed ASIRs per 100000 population were 40.38 (40.14 to 40.62) in men and 19.61 (19.47 to 19.75) in women (Table 1). In this age group, the ASIRs of gastric cancer decreased from 2000–2019 in males and females with AAPCs of -1.58% (-1.77 to -1.38) and -1.16% (-1.34 to -0.97), respectively (Table 1). Also, there was a steady decrease in the incidence rate in all races, with Hispanic men having the lowest AAPC (-2.36%, [-2.65 to -2.05]) (Table 1).

**Younger adults (Ages<55).** Between 2000 and 2019, there were 41327 gastric cancer cases reported in adults <55 years old. Most of the cases were adenocarcinoma (74.41%) (not shown in the table). A total of 57.80% of cases occurred in men and 47.04% occurred in NHWs. The delayed ASIRs per 100000 population were 2.13 (2.11 to 2.16) in men and 1.54 (1.52 to 1.57) in women. From 2000 to 2019, unlike men who experienced no change, the ASIR of gastric cancer in women rose significantly (AAPC: 2.43%; [1.94 to 2.76]). Hispanic women (2.02%; [1.44 to 2.62]) and NHB men (-1.83%; [-2.49 to -1.21]) had the highest and lowest AAPCs, respectively (Table 1).

**Age and sex patterns.** Between 2000 and 2019, across all races and ethnicities, the delay-adjusted incidence rate of gastric cancer showed minimal variation in both males and females within the 10–14 to 20–24 age groups. However, it notably increased in the 75–79 and 85–89 age groups for men and women, respectively (S4A Fig). For Hispanics, the incidence rates of gastric cancer in both males and females exhibited minimal changes from the 15–19 to 20–24

**Table 1. Counts and age-standardized rate of gastric cancer incidence per 100,000 and average annual percent change from 2000 to 2019 in the United States, by age, sex, and race.**

| | All age | Age ≥ 55 | Age < 55 |
|---|---|---|---|
| ***All races/ethnicities*** | | | |
| **Women** | | | |
| Cases (%) | 90094 (39.63) | 72656 (31.96) | 17438 (7.67) |
| Delayed ASIR (95% CI) | 5.4 (5.37 to 5.44) | 19.61 (19.47 to 19.75) | 1.54 (1.52 to 1.57) |
| AAPC (95% CI) | -0.42 (-0.58 to -0.25) | -1.16 (-1.34 to -0.97) | 2.43 (1.94 to 2.76) |
| **Men** | | | |
| Cases (%) | 137245 (60.37) | 113356 (49.86) | 23889 (10.51) |
| Delayed ASIR (95% CI) | 10.3 (10.25 to 10.36) | 40.38 (40.14 to 40.62) | 2.13 (2.11 to 2.16) |
| AAPC (95% CI) | -1.32 (-1.47 to -1.16) | -1.58 (-1.77 to -1.38) | 0.04 (-0.26 to 0.34) |
| ***Hispanic*** | | | |
| **Women** | | | |
| Cases (%) | 19142 (44.32) | 13188 (30.53) | 5954 (13.79) |
| Delayed ASIR (95% CI) | 8.84 (8.71 to 8.97) | 31.8 (31.24 to 32.35) | 2.61 (2.54 to 2.68) |
| AAPC (95% CI) | -0.91 (-1.36 to -0.43) | -1.7 (-2.19 to -1.12) | 2.02 (1.44 to 2.62) |
| **Men** | | | |
| Cases (%) | 24049 (55.68) | 17405 (40.30) | 6644 (15.38) |
| Delayed ASIR (95% CI) | 14.13 (13.93 to 14.32) | 55.51 (54.63 to 56.39) | 2.88 (2.81 to 2.95) |
| AAPC (95% CI) | -2.02 (-2.28 to -1.73) | -2.36 (-2.65 to -2.05) | -0.15 (-0.55 to 0.31) |
| ***NHB*** | | | |
| **Women** | | | |
| Cases (%) | 13921 (43.60) | 11081 (34.70) | 2840 (8.90) |
| Delayed ASIR (95% CI) | 8.55 (8.4 to 8.69) | 32.46 (31.85 to 33.08) | 2.05 (1.98 to 2.13) |
| AAPC (95% CI) | -1.3 (-1.73 to -0.84) | -1.87 (-2.42 to -1.29) | 1.18 (0.46 to 1.92) |
| **Men** | | | |
| Cases (%) | 18010 (56.40) | 14342 (44.91) | 3668 (11.49) |
| Delayed ASIR (95% CI) | 15.83 (15.59 to 16.08) | 63.21 (62.11 to 64.32) | 2.96 (2.87 to 3.06) |
| AAPC (95% CI) | -2.02 (-2.37 to -1.67) | -2.06 (-2.45 to -1.66) | -1.83 (-2.49 to -1.21) |
| ***NHW*** | | | |
| **Women** | | | |
| Cases (%) | 45906 (36.51) | 39699 (31.57) | 6207 (4.94) |
| Delayed ASIR (95% CI) | 3.88 (3.85 to 3.92) | 14.73 (14.58 to 14.87) | 0.94 (0.91 to 0.96) |
| AAPC (95% CI) | -0.82 (-1.09 to -0.55) | -1.42 (-1.77 to -1.08) | 1.79 (1.35 to 2.25) |
| **Men** | | | |
| Cases (%) | 79823 (63.49) | 69058 (54.93) | 10765 (8.56) |
| Delayed ASIR (95% CI) | 8.43 (8.37 to 8.49) | 33.63 (33.37 to 33.88) | 1.59 (1.56 to 1.62) |
| AAPC (95% CI) | -1.58 (-1.74 to -1.46) | -1.85 (-2.07 to -1.7) | -0.06 (-0.5 to 0.37) |

Abbreviations: NHW: Non-Hispanic White; NHB: Non-Hispanic Black; ASIR: Age-standardized incidence rate; CI: Confidence interval, AAPC: Average annual percent change.

age groups. However, a remarkable increase was observed in the 75–79 and 80–84 age groups for men and women, respectively (S4B Fig). In NHBs, the incidence rates of gastric cancer exhibited minimal variations in both males and females between the 15–19 and 25–29 age groups. However, the incidence rates steadily increased for both sexes from the 30–34 to the 75–79 age groups (S4C Fig). Among NHWs, the incidence rates of gastric cancer displayed minimal variations from the 15–19 to the 20–24 age groups. Starting from the 25–29 age group and continuing to the 80–84 age group, both males and females experienced steady

increases in the incidence rates. Notably, among men, there was a notable rise in the incidence rate from the 20–29 age bracket to the 85+ age group, as opposed to women (S4D Fig).

**Effects of COVID-19 on gastric cancer incidence.** The ASIR of gastric cancer was greatly reduced across all races/ethnicities in both sexes within all age groups (percent change (PC): -8.92; 95% CI: -11.18 to -6.67) and those aged ≥55 (PC: -8.72; [-11.26 to -6.17]) and <55 (PC: -9.71; [-14.84 to -4.58]) from 2019 to November 2020. Moreover, there were significant decreases in gastric cancer for males (PC: -8.09; [-11.13 to -5.04]) and females (PC: -10.39; [-14.03 to -6.75]) in all stratifications of age during the COVID-19 pandemic (Table 2).

## Adenocarcinoma

**Overall incidence.** Over 2000–2019, our study found 143803 adenocarcinoma cases in all age groups in the US. A significant portion of cases were men (66.01%), NHWs (64.23%), and aged≥55 years (85.54%) (Table 3 and S5–S7 Figs). The ASIRs per 100000 population were 7.18 (7.13 to 7.22) for men and 2.90 (2.87 to 2.92) for women. The AAPCs for men and women were -1.67% (-1.82 to -1.51) and -1.13% (-1.42 to -0.97), respectively (Table 3 and S5 Fig). NHB men had the highest ASIR (10.67; [10.46 to 10.88]), and NHB women had the lowest AAPC (-2.86%; [-3.43 to -2.32]) (Table 3 and S6 Fig).

**Older adults (Ages≥55).** Overall, 122995 adenocarcinoma cases were recorded over the period 2000–2019 in the US in this group. Most of them were men (65.78%) and NHWs (66.61%). The ASIRs per 100000 population were 29.01 (28.81 to 29.22) in men and 11.37 (11.26 to 11.48) in women. Women had a slightly higher AAPC of gastric adenocarcinoma incidence rate among older adults than men (-1.88% vs. -1.90%) (Table 3). All races had significant declines in the ASIRs, with NHB women having the highest decrease (AAPC: -3.22%; [-3.83 to -2.62]) (Table 3).

**Younger adults (Ages<55).** Overall, 20808 cases of adenocarcinoma were recorded in the US in this age group. The majority of them were men (67.33%) and NHWs (49.88%). The ASIRs per 100000 population were 1.24 (1.22 to 1.27) for men and 0.60 (0.58 to 0.61) for women. Unlike men who had a constant incidence rate (AAPC: -0.21%; [-0.74 to 0.32]), there was an increase in the ASIRs of gastric adenocarcinoma among women from 2000 to 2019 (AAPC: 2.39%; [1.70 to 2.78]) (Table 3). Only Hispanic women experienced an increase in the ASIRs from 2000 to 2019 (AAPC: 3.07%; [1.73 to 3.97]) (Table 3).

## Signet ring carcinoma

**Overall incidence.** A total of 36121 cases of signet ring carcinoma were reported in the US. The majority of them were men (52.22%), NHWs (57.54%), and aged≥55 years (70.59%) (Table 4 and S8–S10 Figs). The ASIRs per 100000 population were 1.37 (1.35 to 1.39) and 1.06 (1.05 to 1.08) for men and women, respectively. There was a higher rate of decline in the ASIRs of signet ring carcinoma among men (AAPC: -3.85%; [-4.49 to -3.38]) than women (AAPC: -2.25%; [-2.54 to -1.99]) (Table 4 and S8 Fig). NHB men had the greatest decline in the ASIR of signet ring carcinoma with an AAPC of -5.46% (-7.02 to -4.35) (Table 4).

**Older adults (Ages≥55).** Over 2000–2019, a total of 25497 cases of signet ring carcinoma among older adults in the US were reported. The majority of them were men (53.38%) and NHWs (64.86%). The ASIRs per 100000 population were 4.68 (4.60 to 4.76) for men and 3.21 (3.15 to 3.27) for women. Men had a greater decline than women from 2000–2019 (AAPC: -4.29%; [-4.94 to -3.77] vs. AAPC: -3.52%; [-4.15 to -3.01]). NHB men had the highest rate of decline (AAPC: -5.23%; [-6.76 to -3.90]) (Table 4).

**Younger adults (Ages<55).** There were 10624 cases of signet ring carcinoma among young adults between 2000 and 2019. Most were women (50.55%) and Hispanics (44.92%).

**Table 2. Percent change in age-standardized and delay-adjusted incidence rates of gastric cancer from 2019 to 2020, by age, race and sex in the United States, using the November 2022 data submission.**

| Races/ethnicities | Sex | Age | 2019 Delayed ASIR (95% CI) | 2020 Delayed ASIR (95% CI) | PC (95% CI) |
|---|---|---|---|---|---|
| All | Both | All | 7.06 (6.94 to 7.19) | 6.43 (6.31 to 6.55) | -8.92 (-11.18 to -6.67) |
| All | Both | ≥55 | 25.47 (24.98 to 25.97) | 23.25 (22.78 to 23.73) | -8.72 (-11.26 to -6.17) |
| All | Both | <55 | 2.06 (1.98 to 2.15) | 1.86 (1.78 to 1.95) | -9.71 (-14.84 to -4.58) |
| All | Male | All | 9.15 (8.94 to 9.37) | 8.41 (8.21 to 8.62) | -8.09 (-11.13 to -5.04) |
| All | Male | <55 | 2.11 (1.99 to 2.23) | 1.93 (1.82 to 2.05) | -8.53 (-16.08 to -0.98) |
| All | Male | ≥55 | 35.09 (34.21 to 35.98) | 32.27 (31.44 to 33.12) | -8.04 (-11.37 to -4.7) |
| All | Female | All | 5.39 (5.24 to 5.54) | 4.83 (4.69 to 4.98) | -10.39 (-14.03 to -6.75) |
| All | Female | <55 | 2.02 (1.9 to 2.14) | 1.79 (1.68 to 1.91) | -11.39 (-19.16 to -3.61) |
| All | Female | ≥55 | 17.8 (17.25 to 18.36) | 16.01 (15.49 to 16.55) | -10.06 (-14.12 to -5.99) |
| Hispanic | Both | All | 10.14 (9.76 to 10.53) | 8.99 (8.64 to 9.36) | -11.34 (-16.23 to -6.46) |
| Hispanic | Both | <55 | 3.2 (2.99 to 3.41) | 2.77 (2.58 to 2.97) | -13.44 (-21.9 to -4.98) |
| Hispanic | Both | ≥55 | 35.7 (34.1 to 37.35) | 31.9 (30.41 to 33.45) | -10.64 (-16.51 to -4.77) |
| Hispanic | Female | all | 8.76 (8.29 to 9.25) | 7.73 (7.30 to 8.19) | -11.76 (-18.75 to -4.76) |
| Hispanic | Female | <55 | 3.47 (3.16 to 3.79) | 2.93 (2.65 to 3.23) | -15.56 (-26.96 to -4.16) |
| Hispanic | Female | ≥55 | 28.25 (26.38 to 30.23) | 25.41 (23.66 to 27.25) | -10.05 (-18.8 to -1.31) |
| Hispanic | Male | All | 12.01 (11.37 to 12.66) | 10.69 (10.10 to 11.30) | -10.99 (-17.96 to -4.02) |
| Hispanic | Male | <55 | 2.94 (2.66 o 3.23) | 2.62 (2.35 to 2.90) | -10.88 (-23.39 to 1.62) |
| Hispanic | Male | ≥55 | 45.39 (42.62 to 48.28) | 40.41 (37.83 to 43.11) | -10.97 (-18.95 to -2.99) |
| NHB | Both | All | 9.95 (9.48 to 10.44) | 9.13 (8.68 to 9.59) | -8.24 (-14.51 to -1.97) |
| NHB | Both | <55 | 2.47 (2.21 to 2.75) | 2.1 (1.86 to 2.37) | -14.98 (-28.97 to -0.99) |
| NHB | Both | ≥55 | 37.51 (35.55 to 39.55) | 35 (33.12 to 36.96) | -6.69 (-13.79 to 0.41) |
| NHB | Female | All | 7.89 (7.35 to 8.47) | 7.23 (6.71 to 7.78) | -8.37 (-17.62 to 0.89) |
| NHB | Female | <55 | 2.37 (2.02 to 2.76) | 2.09 (1.76 to 2.46) | -11.81 (-32.15 to 8.52) |
| NHB | Female | ≥55 | 28.22 (26.04 to 30.54) | 26.14 (24.07 to 28.34) | -7.37 (-17.86 to 3.12) |
| NHB | Male | All | 12.99 (12.14 to 13.88) | 11.88 (11.08 to 12.72) | -8.55 (-17.32 to 0.23) |
| NHB | Male | <55 | 2.58 (2.019 to 3.00) | 2.12 (1.77 to 2.52) | -17.83 (-36.91 to 1.25) |
| NHB | Male | ≥55 | 51.32 (47.63 to 55.21) | 47.80 (44.29 to 51.51) | -6.86 (-16.6 to 2.89) |
| NHW | Both | All | 5.35 (5.21 to 5.49) | 5.00 (4.87 to 5.14) | -6.54 (-10.05 to -3.03) |
| NHW | Both | <55 | 1.34 (1.25 to 1.44) | 1.32 (1.22 to 1.42) | -1.49 (-11.76 to 8.77) |
| NHW | Both | ≥55 | 20.10 (19.57 to 20.64) | 18.57 (18.06 to 19.09) | -7.61 (-11.13 to -4.1) |
| NHW | Female | All | 3.63 (3.48 to 3.80) | 3.31 (3.16 to 3.47) | -8.82 (-14.66 to -2.97) |
| NHW | Female | <55 | 1.15 (1.03 to 1.28) | 1.11 (0.99 to 1.25) | -3.48 (-17.69 to 10.73) |
| NHW | Female | ≥55 | 12.78 (12.21 to 13.37) | 11.41 (10.87 to 11.97) | -10.72 (-16.57 to -4.87) |
| NHW | Male | All | 7.39 (7.16 to 7.63) | 7.02 (6.79 to 7.25) | -5.01 (-9.4 to -0.62) |
| NHW | Male | <55 | 1.53 (1.39 to 1.68) | 1.53 (1.38 to 1.38) | 0 (-12.68 to 12.68) |
| NHW | Male | ≥55 | 28.97 (28.02 to 29.94) | 27.24 (26.32 to 28.19) | -5.97 (-10.47 to -1.47) |

Abbreviations: NHW: Non-Hispanic White; NHB: Non-Hispanic Black; ASIR: Age-standardized Incidence rate; CI: Confidence interval, PC: percent change.

The ASIRs per 100000 population were 0.47 (0.46 to 0.49) for men and 0.48 (0.47 to 0.49) for women. NHB men had the highest decline among other groups from 2000–2019 (AAPC: -6.05%; [-8.54 to -4.51]) (Table 4).

## Carcinoid tumor

**Overall incidence.** We identified 13279 cases of carcinoid tumors in all age groups in the US from 2000 to 2019. They were mostly among women (61.08%), NHWs (61.67%), and

**Table 3. Counts and age-standardized rate of gastric adenocarcinoma incidence per 100,000 and average annual percent change from 2000 to 2019 in the United States, by age, sex, and race.**

| | *All age* | *Age ≥ 55* | *Age < 55* |
|---|---|---|---|
| *All races/ethnicities* | | | |
| **Women** | | | |
| Cases (%) | 48880 (33.99) | 42084 (29.27) | 6796 (4.72) |
| ASIR (95% CI) | 2.9 (2.87 to 2.92) | 11.37 (11.26 to 11.48) | 0.6 (0.58 to 0.61) |
| AAPC (95% CI) | -1.13 (-1.42 to -0.97) | -1.88 (-2.13 to -1.74) | 2.39 (1.7 to 2.78) |
| **Men** | | | |
| Cases (%) | 94923 (66.01) | 80911 (56.27) | 14012 (9.74) |
| ASIR (95% CI) | 7.18 (7.13 to 7.22) | 29.01 (28.81 to 29.22) | 1.24 (1.22 to 1.27) |
| AAPC (95% CI) | -1.67 (-1.82 to -1.51) | -1.9 (-2.09 to -1.71) | -0.21 (-0.74 to 0.32) |
| *Hispanic* | | | |
| **Women** | | | |
| Cases (%) | 9879 (38.27) | 7578 (29.36) | 2301 (8.91) |
| ASIR (95% CI) | 4.79 (4.69 to 4.89) | 18.7 (18.27 to 19.13) | 1.01 (0.97 to 1.05) |
| AAPC (95% CI) | -1.51 (-2.17 to -1.07) | -2.72 (-3.21 to -2.17) | 3.07 (1.73 to 3.97) |
| **Men** | | | |
| Cases (%) | 15936 (61.73) | 12281 (47.57) | 3655 (14.16) |
| ASIR (95% CI) | 9.74 (9.58 to 9.91) | 39.71 (38.97 to 40.47) | 1.6 (1.55 to 1.65) |
| AAPC (95% CI) | -2.36 (-2.7 to -1.99) | -2.68 (-3.02 to -2.31) | -0.16 (-1.31 to 1.17) |
| *NHB* | | | |
| **Women** | | | |
| Cases (%) | 7521 (38.92) | 6457 (33.41) | 1064 (5.51) |
| ASIR (95% CI) | 4.72 (4.61 to 4.83) | 19.29 (18.81 to 19.77) | 0.76 (0.72 to 0.81) |
| AAPC (95% CI) | -2.86 (-3.43 to -2.32) | -3.22 (-3.83 to -2.62) | -0.42 (-1.46 to 0.63) |
| **Men** | | | |
| Cases (%) | 11802 (61.08) | 9797 (50.70) | 2005 (10.38) |
| ASIR (95% CI) | 10.67 (10.46 to 10.88) | 44.01 (43.08 to 44.94) | 1.61 (1.54 to 1.68) |
| AAPC (95% CI) | -2.25 (-2.96 to -1.53) | -2.25 (-2.85 to -1.61) | -2.22 (-3.12 to 1.32) |
| *NHW* | | | |
| **Women** | | | |
| Cases (%) | 24960 (30.79) | 22608 (27.89) | 2352 (2.90) |
| ASIR (95% CI) | 2.05 (2.03 to 2.08) | 8.33 (8.22 to 8.44) | 0.35 (0.33 to 0.36) |
| AAPC (95% CI) | -1.83 (-2.29 to -1.53) | -2.22 (-2.76 to -1.87) | 0.15 (-0.75 to 1.02) |
| **Men** | | | |
| Cases (%) | 56097 (69.21) | 49438 (60.99) | 6659 (8.22) |
| ASIR (95% CI) | 5.92 (5.87 to 5.97) | 24.15 (23.93 to 24.37) | 0.97 (0.95 to 1) |
| AAPC (95% CI) | -1.8 (-2.02 to -1.53) | -1.99 (-2.21 to -1.81) | -0.28 (-0.85 to 0.26) |

Abbreviations: NHW: Non-Hispanic White; NHB: Non-Hispanic Black; ASIR: Age-standardized incidence rate; CI: Confidence interval, AAPC: Average annual percent change.

aged≥55 years (71.33%) (Table 5 and S11–S13 Figs). The ASIRs per 100000 population were 0.37 (0.36 to 0.38) for men and 0.50 (0.49 to 0.51) for women. Women experienced a higher increase in the incidence rates (AAPC: 4.53%; [3.68 to 6.23]) compared to men (AAPC: 3.04%; [2.43 to 3.84]) (Table 5 and S11 Fig). There was an increase in all study groups, with NHB men having the highest increase (AAPC: 5.26%; [2.79 to 9.69]) (Table 5 and S12 Fig).

**Older adults (Ages≥55).** Overall, 9472 cases were identified in the US from 2000–2019 in this age group, with the majority being women (58.93%) and NHWs (66.76%). The ASIRs per

**Table 4. Counts and age-standardized rate of signet ring carcinoma incidence per 100,000 and average annual percent change from 2000 to 2019 in the United States, by age, sex, and race.**

| | All age | Age ≥ 55 | Age < 55 |
|---|---|---|---|
| *All races/ethnicities* | | | |
| **Women** | | | |
| Cases (%) | 17257 (47.78) | 11886 (32.91) | 5371 (14.87) |
| ASIR (95% CI) | 1.06 (1.05 to 1.08) | 3.21 (3.15 to 3.27) | 0.48 (0.47 to 0.49) |
| AAPC (95% CI) | -2.25 (-2.54 to -1.99) | -3.52 (-4.15 to -3.01) | 0.06 (-0.44 to 0.53) |
| **Men** | | | |
| Cases (%) | 18864 (52.22) | 13611 (37.68) | 5253 (14.54) |
| ASIR (95% CI) | 1.37 (1.35 to 1.39) | 4.68 (4.6 to 4.76) | 0.47 (0.46 to 0.49) |
| AAPC | -3.85 (-4.49 to -3.38) | -4.29 (-4.94 to -3.77) | -2.37 (-3.58 to -1.42) |
| *Hispanic* | | | |
| **Women** | | | |
| Cases (%) | 4616 (51.11) | 2497 (27.65) | 2119 (23.46) |
| ASIR (95% CI) | 1.94 (1.88 to 1.99) | 5.69 (5.46 to 5.92) | 0.92 (0.88 to 0.96) |
| AAPC (95% CI) | -2.78 (-3.69 to -1.81) | -4.15 (-5.79 to -2.5) | -0.56 (-1.33 to 0.28) |
| **Men** | | | |
| Cases (%) | 4415 (48.89) | 2475 (27.41) | 1940 (21.48) |
| ASIR (95% CI) | 2.18 (2.11 to 2.26) | 7.2 (6.9 to 7.51) | 0.82 (0.79 to 0.86) |
| AAPC (95% CI) | -3.95 (-4.84 to -2.58) | -3.98 (-4.81 to -3.13) | -1.39 (-2.61 to -0.05) |
| *NHB* | | | |
| **Women** | | | |
| Cases (%) | 2048 (47.68) | 1356 (31.57) | 692 (16.11) |
| ASIR (95% CI) | 1.21 (1.16 to 1.27) | 3.83 (3.62 to 4.04) | 0.5 (0.47 to 0.54) |
| AAPC (95% CI) | -2.69 (-3.68 to -1.73) | -4.02 (-5.52 to -2.67) | -0.5 (-6.1 to 3.55) |
| **Men** | | | |
| Cases (%) | 2247 (52.32) | 1524 (35.48) | 723 (16.84) |
| ASIR (95% CI) | 1.79 (1.71 to 1.87) | 6.22 (5.89 to 6.56 | 0.59 (0.55 to 0.63) |
| AAPC (95% CI) | -5.46 (-7.02 to -4.35) | -5.23 (-6.76 to -3.9) | -6.05 (-8.54 to -4.51) |
| *NHW* | | | |
| **Women** | | | |
| Cases (%) | 8006 (44.33) | 6374 (35.30) | 1632 (9.04) |
| ASIR (95% CI) | 0.71 (0.69 to 0.72) | 2.39 (2.34 to 2.45) | 0.25 (0.24 to 0.26) |
| AAPC (95% CI) | -2.48 (-3 to -1.99) | -4.17 (-4.94 to -3.18) | 0.05 (-1.07 to 1.14) |
| **Men** | | | |
| Cases (%) | 10053 (55.67) | 8124 (44.98) | 1929 (10.68) |
| ASIR (95% CI) | 1.05 (1.03 to 1.08) | 3.88 (3.79 to 3.96) | 0.29 (0.28 to 0.3) |
| AAPC (95% CI) | -4.5 (-5.6 to -3.71) | -4.48 (-5.55 to -3.66) | -1.33 (-2.55 to -0.24) |

Abbreviations: NHW: Non-Hispanic White; NHB: Non-Hispanic Black; ASIR: Age-standardized incidence rate; CI: Confidence interval, AAPC: Average annual percent change.

100000 population were 1.32 (1.27 to 1.36) for men and 1.51 (1.47 to 1.55) for women. Women had a higher increase in incidence rate (AAPC: 3.70%; [3.12 to 4.50]) from 2000 to 2019 compared to men (AAPC: 2.93%; [2.23 to 3.79]). NHB men had the highest increase in the incidence rate from 2000 to 2019 (AAPC: 5.81%; [1.88 to 11.71]) (Table 5).

**Younger adults (Ages<55).** Over the 2000 and 2019 period, there were an overall 3807 cases of carcinoid tumors in young adults in the US. They mostly occurred in women (66.45%) and NHWs (48.82%). The ASIRs per 100000 population were 0.11 (0.11 to 0.12) for

**Table 5. Counts and age-standardized rate of carcinoid tumor incidence per 100,000 and average annual percent change from 2000 to 2019 in the United States, by age, sex, and race.**

|  | *All age* | *Age ≥ 55* | *Age < 55* |
|---|---|---|---|
| *All races/ethnicities* | | | |
| **Women** | | | |
| Cases (%) | 8112 (61.08) | 5582 (42.04) | 2530 (19.05) |
| ASIR (95% CI) | 0.5 (0.49 to 0.51) | 1.51 (1.47 to 1.55) | 0.22 (0.21 to 0.23) |
| AAPC (95% CI) | 4.53 (3.68 to 6.23) | 3.7 (3.12 to 4.5) | 6.86 (4.68 to 9.24) |
| **Men** | | | |
| Cases (%) | 5167 (38.92) | 3890 (29.24) | 1277 (9.62) |
| ASIR (95% CI) | 0.37 (0.36 to 0.38) | 1.32 (1.27 to 1.36) | 0.11 (0.11 to 0.12) |
| AAPC (95% CI) | 3.04 (2.43 to 3.84) | 2.93 (2.23 to 3.79) | 2.97 (1.27 to 4.55) |
| *Hispanic* | | | |
| **Women** | | | |
| Cases (%) | 2189 (70.02) | 1255 (40.15) | 934 (29.88) |
| ASIR (95% CI) | 0.91 (0.88 to 0.95) | 2.76 (2.61 to 2.92) | 0.41 (0.39 to 0.44) |
| AAPC (95% CI) | 4.93 (3.93 to 6.35) | 4.26 (2.6 to 6.77) | 5.85 (4.79 to 7.27) |
| **Men** | | | |
| Cases (%) | 937 (29.98) | 611 (19.54) | 326 (10.43) |
| ASIR (95% CI) | 0.49 (0.46 to 0.52) | 1.77 (1.62 to 1.92) | 0.14 (0.13 to 0.16) |
| AAPC (95% CI) | 1.84 (0.38 to 3.7) | 1.13 (-0.49 to 3.21) | 4.22 (0.96 to 9.17) |
| *NHB* | | | |
| **Women** | | | |
| Cases (%) | 1034 (63.13) | 675 (41.21) | 359 (21.92) |
| ASIR (95% CI) | 0.59 (0.56 to 0.63) | 1.82 (1.69 to 1.97) | 0.26 (0.23 to 0.29) |
| AAPC (95% CI) | 3.84 (2.74 to 5.19) | 2.04 (0.64 to 3.82) | 7.06 (5.56 to 9.13) |
| **Men** | | | |
| Cases (%) | 604 (36.87) | 421 (25.70) | 183 (11.17) |
| ASIR (95% CI) | 0.47 (0.43 to 0.51) | 1.65 (1.49 to 1.83) | 0.15 (0.13 to 0.17) |
| AAPC (95% CI) | 5.26 (2.79 to 9.69) | 5.81 (1.88 to 11.71) | 2.49 (-4.16 to 10.13) |
| *NHW* | | | |
| **Women** | | | |
| Cases (%) | 4453 (58.07) | 3369 (43.94) | 1084 (14.13) |
| ASIR (95% CI) | 0.41 (0.39 to 0.42) | 1.3 (1.26 to 1.35) | 0.16 (0.15 to 0.17) |
| AAPC (95% CI) | 4.03 (3.38 to 4.78) | 3.15 (2.57 to 3.88) | 5.4 (3.66 to 7.44) |
| **Men** | | | |
| Cases (%) | 3215 (41.93) | 2580 (33.65) | 635 (8.28) |
| ASIR (95% CI) | 0.33 (0.32 to 0.35) | 1.21 (1.17 to 1.26) | 0.09 (0.09 to 0.1) |
| AAPC (95% CI) | 3.45 (2.63 to 4.39) | 3.09 (2.16 to 4.25) | 2.86 (1.07 to 4.11) |

Abbreviations: NHW: Non-Hispanic White; NHB: Non-Hispanic Black; ASIR: Age-standardized incidence rate; CI: Confidence interval, AAPC: Average annual percent change.

men and 0.22 (0.21 to 0.23) for women. Unlike NHB men, who experienced no change in incidence rate, other groups showed an increase, with NHB women having the highest increase in incidence rate from 2000 to 2019 (AAPC: 7.06%; [5.56 to 9.13]) (Table 5).

## GIST

**Overall incidence.** Between 2000 and 2019, 15273 cases of GIST were recorded in all age groups in the US. They mostly reported in women (50.56%), NHWs (59.66%), and those ≥55 years (78.07%) (Table 6 and S14–S16 Figs). The ASIRs per 100000 population were 0.54 (0.53 to 0.56) for men and 0.47 (0.46 to 0.48) for women. The AAPCs for men and women were 6.34% (5.19 to 7.59) and 8.02% (5.13 to 10.83), respectively (Table 6 and S14 Fig). NHB men

**Table 6. Counts and age-standardized rate of gastrointestinal stromal tumor incidence per 100,000 and average annual percent change from 2000 to 2019 in the United States, by age, sex, and race.**

| | *All age* | *Age ≥ 55* | *Age < 55* |
|---|---|---|---|
| *All races/ethnicities* | | | |
| **Women** | | | |
| Cases (%) | 7722 (50.56) | 6060 (39.68) | 1662 (10.88) |
| ASIR (95% CI) | 0.47 (0.46 to 0.48) | 1.65 (1.61 to 1.69) | 0.15 (0.14 to 0.15) |
| AAPC (95% CI) | 8.02 (5.13 to 10.83) | 7.65 (6.06 to 9.43) | 5.37 (3.96 to 6.8) |
| **Men** | | | |
| Cases (%) | 7551 (49.44) | 5864 (38.39) | 1687 (11.05) |
| ASIR (95% CI) | 0.54 (0.53 to 0.56) | 2 (1.95 to 2.05) | 0.15 (0.14 to 0.16) |
| AAPC (95% CI) | 6.34 (5.19 to 7.59) | 7.08 (5.27 to 9.52) | 5.83 (3.14 to 7.87) |
| *Hispanic* | | | |
| **Women** | | | |
| Cases (%) | 934 (51.63) | 662 (36.59) | 272 (15.04) |
| ASIR (95% CI) | 0.42 (0.39 to 0.45) | 1.53 (1.41 to 1.65) | 0.12 (0.1 to 0.13) |
| AAPC (95% CI) | 9.51 (6.19 to 13.56) | 10.54 (7.67 to 16.28) | 5.41 (3.07 to 9.06) |
| **Men** | | | |
| Cases (%) | 875 (48.37) | 611 (33.78) | 264 (14.59) |
| ASIR (95% CI) | 0.47 (0.44 to 0.5) | 1.78 (1.63 to 1.94) | 0.11 (0.1 to 0.13) |
| AAPC (95% CI) | 7.99 (6.13 to 11.5) | 8.89 (4.62 to 16.41) | 3.03 (-0.01 to 7.21) |
| *NHB* | | | |
| **Women** | | | |
| Cases (%) | 1998 (53.68) | 1494 (40.14) | 504 (13.54) |
| ASIR (95% CI) | 1.18 (1.13 to 1.23) | 4.18 (3.97 to 4.4) | 0.36 (0.33 to 0.39) |
| AAPC (95% CI) | 6.43 (5.13 to 8.29) | 6.15 (4.47 to 10.93) | 5.99 (3.57 to 9.31) |
| **Men** | | | |
| Cases (%) | 1724 (46.32) | 1272 (34.18) | 452 (12.14) |
| ASIR (95% CI) | 1.39 (1.32 to 1.46) | 5.17 (4.87 to 5.48) | 0.36 (0.33 to 0.4) |
| AAPC (95% CI) | 5.39 (3.48 to 8.12) | 5.13 (3.33 to 8.62) | 4.57 (2.8 to 6.76) |
| *NHW* | | | |
| **Women** | | | |
| Cases (%) | 3960 (48.39) | 3249 (39.70) | 711 (8.69) |
| ASIR (95% CI) | 0.35 (0.34 to 0.36) | 1.24 (1.2 to 1.29) | 0.11 (0.1 to 0.12) |
| AAPC (95% CI) | 7.35 (3.89 to 10.62) | 7.35 (3.9 to 10.79) | 8.35 (4.52 to 10.93) |
| **Men** | | | |
| Cases (%) | 4223 (51.61) | 3423 (41.83) | 800 (9.78) |
| ASIR (95% CI) | 0.44 (0.43 to 0.45) | 1.62 (1.56 to 1.67) | 0.12 (0.11 to 0.13) |
| AAPC (95% CI) | 6.54 (5.61 to 8.24) | 6.75 (4.81 to 8.52) | 4.99 (2.37 to 7.63) |

Abbreviations: NHW: Non-Hispanic White; NHB: Non-Hispanic Black; ASIR: Age-standardized Incidence rate; CI: Confidence interval, AAPC: Average annual percent change.

had the highest ASIR (1.39; [1.32 to 1.46]), and Hispanic women had the highest AAPC (9.51%; [6.19 to 13.56]) (Table 6 and S15 Fig).

**Older adults (Ages≥55).** Overall, there were 11924 cases of GIST reported between 2000 and 2019 in this age group. It was mostly occurred in women (50.82%) and NHWs (62.29%). The ASIRs per 100000 population were 2.00 (1.95 to 2.05) in men and 1.65 (1.61 to 1.69) in women. Women had a higher AAPC for GIST incidence rate among older adults than men (7.65% vs. 7.08%) (Table 6). Over 2000–2019, it was significantly increased in all races/ethnicities, and Hispanic women had the largest one (AAPC: 10.54%; [7.67 to 16.28]) (Table 6).

**Younger adults (Ages<55).** From 2000 to 2019, GIST comprised of 3349 cases in the US in this age group. Most of them were men (50.37%) and NHWs (50.31%). The ASIRs per 100000 population were 0.15 (0.14 to 0.16) for men and 0.15 (0.14 to 0.15) for women. The AAPCs were higher in men than in women (5.83% vs. 5.37%) (Table 6). Unlike Hispanic men, who showed no significant changes, other groups experienced increases, with NHW women having the highest increase in the ASIRs (AAPC: 8.35%; [4.52 to 10.93]) (Table 6).

## Discussion

Our findings showed that the ASIR of gastric cancer has decreased in the US over the last two decades, although there are still a high number of incident cases in the US in 2020. Also, gastric cancer had more incident cases among the elderly, males, and NHWs. Gastric adenocarcinoma accounted for the highest number of cases of gastric cancer. Furthermore, COVID-19 led to a significant decrease in the ASIRs of gastric cancer in the US. This decrease may be attributed to multiple factors related to the COVID-19 pandemic. These factors include disruptions in healthcare services, such as reduced access to routine screenings and diagnostic procedures, delays in seeking medical attention due to lockdowns, and fear of being affected by the virus. Additionally, reporting issues and temporary reallocation of healthcare resources towards managing the pandemic may have contributed to the apparent decline in reported cases.

Parallel to the previous studies [7, 35], a significant decrease of ASIR was seen in both sexes over 2000–2019 (AAPC of -1.32% and -0.42% for men and women, respectively), thanks to the efficient *Helicobacter pylori* (*H. pylori*) eradication protocol and life-style related risk factors' adjustments at the primary level of prevention [36]. In this regard, a nationwide study on the US population showed a decreasing trend for *H. pylori* from 1999–2018 [37], which may explain the decreasing incidence of gastric cancer in men and women in our study. They also found the most considerable *H. pylori* infection rate in NHBs (40.2%) and Hispanics (36.7%) in the US [37]. In accordance with the findings, NHBs and Hispanics in our study had higher ASIRs of gastric cancer than NHWs. Moreover, the global prevalence of *H. pylori* infection has decreased by about 16% between 1980 and 2022 [38]. Results of a meta-analysis also confirmed a strong significant association between *H. pylori* infection and risk of non-cardia gastric cancer in European and North American populations (odds ratio = 5.37; 95% CI: 4.39, 6.57) [39]. Although the US has a low prevalence of *H. pylori* infection, the antimicrobial resistance rate has recently increased [40]. There are also disparities between the management of *H. pylori* as outlined in the guidelines and its actual implementation in real-world settings [41]. This highlights the importance of improving the healthcare system for early diagnosis and management of *H. pylori* to prevent its consequences, especially gastric cancer. Despite the lower incidence of gastric cancer in the US compared to the global values, its five-year survival rate is about 36% [4, 42, 43]. The existence of efficient prevention by early diagnosis of *H. pylori* [7, 35] has turned a spotlight on this so-called "neglected cancer" [44]. However, the SEER program only provides data on cancer incidence and survival in the US, and risk factors are not reported in the SEER data, which limits us from reevaluating this hypothesis using the current study.

There are several other types of risk factors for gastric cancer, which can be categorized into lifestyle (e.g., smoking and alcohol consumption), environmental (e.g., ionizing radiation and unhealthy diet), occupational (e.g., mineral dust and cement exposure), and socioeconomic (e.g., low income and education levels), in addition to genetic factors and family history of gastric cancer [45]. Lifestyle factors, such as smoking and alcohol consumption, have been linked to an increased risk of gastric cancer. For instance, the age-standardized prevalence rates of tobacco smoking have significantly decreased by 38.7% and 29.8% among females and males in the US from 1990–2019, respectively [46]. This can be one of the potential reasons for a decreased trend of gastric cancer over 2000–2019 in our study. Dietary factors are also significant, with gastric cancer being the third most common cancer related to diet in the US, contributing to 6.8% of the population-attributable fractions of cancers related to dietary habits [47]. Unhealthy diets, particularly those high in salted, smoked, or pickled foods, can increase the risk of gastric cancer. Environmental factors include exposure to harmful substances such as ionizing radiation. It has also been shown that occupational exposure, like exposures to wood dust and aromatic amine in jobs like agricultural workers, miners, and machine-tool operators can increase the risk of gastric cancer [48]. Given the industrial nature of the US, it is crucial to implement preventive measures and enhance workplace safety to mitigate these risks. Socioeconomic factors significantly impact cancer incidence and outcomes. Generally, the burden of gastric cancer is higher in countries with low levels of socioeconomic status [49]. This also explains the low incidence rates of gastric cancer in the US as a high-income country. Regarding genetic factors, most genetic alterations linked to gastric cancer are acquired, stemming from chromosomal instability, microsatellite instability, alterations in microRNA expression, somatic mutations, or functional single nucleotide polymorphisms [50]. As a result, advances in personalized medicine through targeted therapy holds promise for gastric cancer diagnosis and treatment, with molecular profiling as the cornerstone. Accordingly, identifying a comprehensive array of genetic risk factors tailored to patients' ethnicities is anticipated to become increasingly streamlined and affordable. Large-scale genome-wide association studies will continue to show genetic risk markers that could be integrated into forthcoming prospective studies aimed at stratifying risk within specific populations [50]. Overall, a multifaceted approach encompassing lifestyle modifications, environmental and occupational safety measures, socioeconomic improvements, and advancements in genetic research is essential for effectively managing and reducing the incidence of gastric cancer [51].

Our results showed that most incident cases of gastric cancer were among males aged ≥55 years and NHWs. Ethnic disparities have often led to diversity in clinical aspects of the population. Additionally, migration also plays a vital role in the incidence and survival of gastric cancer [52]. This difference also reflected comparable environmental and genetic factors and various barriers to accessing health services among races living in the US [43, 53]. Migration patterns also significantly influence the incidence and survival rates of gastric cancer. Nevertheless, similarities such as the increasing trend of ASIRs by age were demonstrated in all ethnic groups. Previous studies also showed that almost all races had a similar gradual increase in gastric cancer incidence for both sexes, followed by a higher number of incident cases among males over 40–70 years. Afterward, an expected slight decline was recorded after the 80s [7, 35]. Overall, while ethnic disparities in gastric cancer incidence and survival are evident, the common trends observed in relation to age and other factors highlight the need for targeted interventions that address both universal and population-specific risk factors. Understanding these patterns is crucial for developing effective public health strategies and ensuring equitable access to gastric cancer prevention, diagnosis, and treatment services.

The ASIRs among younger adults of both sexes were not high compared to the elderly. However, the significant positive AAPCs among younger adults, especially young women

(AAPC: 2.43%; 95% CI: 1.94, 2.76) should be considered. In recent studies, there was a rise in gastric cancer incidence, which was considerable among younger Hispanic females [54–56]. Previous studies also examined the effects of genetic factors on different ethnicities/races. However, there is still ambiguity on the real cause of this preference due to inner massive divergence [57, 58]. Also, the birthplace might be a confounding factor not considered in SEER data, which could impact the incidence and characteristics of gastric cancer. For instance, in a study by Chang et al. [56] a significant rise in diffuse gastric cancer was seen in foreign-born Hispanic males and US-born Hispanic females [56]. Generally, while the absolute risk of gastric cancer among younger individuals remains lower than in the elderly, the upward trend in incidence rates among younger adults—especially young women—highlights the need for targeted research and preventive strategies to address this emerging trend.

Merchant and colleagues used SEER data to report the incidence trends of advanced gastric cancer between 1992 and 2011, which showed increasing rates of gastric cancer in Hispanic males [59]. However, our analysis of recent data showed that gastric cancer incidence decreased by about two percent over 2000–2019. There are several reasons for the differences. The mentioned study used data between 1992 and 2011 [59], while we used data from 2000 to 2019. Also, they only reported the burden of advanced gastric cancer [59], whereas our study reported all subtypes. Health committees should pay more attention to young adults in their future active screening policies and healthcare provisions. Moreover, those with large Hispanic populations, such as California and Texas, should take prompt action regarding more cost-effective gastric cancer screenings [54]. Further studies on young adults of different races/ethnicities might help identify the relevant hereditary risk factors for gastric cancer. Inherent risk factors and likely genomic differences across ethnicities, rather than chronic environmental carcinogens, may play a prominent role in the risk of developing gastric cancer [7]. It is also hypothesized that the widespread use of endoscopic screening and detection bias leads to the rise of gastric cancer incidence in some subdivisions. However, this could be merely attributable to local stages of gastric cancer [16, 17]. Unlike adenocarcinoma and signet ring cell carcinoma with negative AAPC, there were remarkable increases in the ASIRs of unfamiliar and rare carcinoid tumors and GISTs. Carcinoids and GISTs significantly increased in almost all age groups and ethnic subgroups, which aligned with findings by Rustgi et al. [17]. With higher AAPC for women, both were primarily seen in females, NHWs, and adults aged at least 55 years. However, AAPC was higher for younger adults, which is likely responsible for the increasing trend of total gastric cancers along with a rise of gastric adenocarcinoma in young Hispanic females due to endoscopic-related detection bias. Furthermore, the increase in proton pump inhibitor utilization since the 2000s [60], has led to more cases of carcinoid tumors being reported. The carcinoid tumor type 1 is also associated with autoimmune gastritis, coincidentally a risk factor for gastric adenocarcinoma [61]. Autoimmune gastritis, with a prevalence of 0.5%–2.0% in the general population, has also increased recently [62]. Similar to the study's findings by Anderson et al. [63], we also found a rising trend of gastric cancer in younger adults. In this regard, Montminy et al. [64] demonstrated that almost 20% of low-grade colorectal and nearly 35% of low-grade rectal cancers were carcinoids. It might highlight the need to design further studies on carcinoid tumors and their probable risk factors so that this recent increase in gastric cancers among youths can be better understood.

Regarding the effects of COVID-19 on the incidence of gastric cancer, there were decreasing trends for gastric cancer in the US, irrespective of histology, age, and race/ethnicity. Recent studies conducted in Korea have revealed that the rate of gastric cancer screening has declined dramatically, leading to the probable risk of delayed diagnosis and treatment [65, 66]. For instance, a Korean study showed a dramatically declined screening rate from 61.9% to 54.6% between 2019 and 2020 [66]. Moreover, a study from the Upper Gastrointestinal Surgical

Society highlighted the massive decrease in surgical and invasive services for patients with gastric cancer and endoscopic screenings [9]. This unfavorable condition might lead to this deceptive negative AAPC in 2020 and to unwillingly late-diagnosed cases of gastric cancer in the following years [9, 67–70]. From another point of view, a Mendelian randomization-based study proposed a causality between COVID-19 and 33 types of cancers, including gastric cancer [71]. Critically ill COVID-19 patients were suggested to be at increased risk of gastric cancer (odds ratio = 1.24 and p-value = 0.03), which might be due to genetic liabilities [71]. Furthermore, severe acute respiratory syndrome coronavirus 2 might bind to angiotensin-converting enzyme 2 (ACE-2) receptors, affecting the gut and its microbiome [72]. This phenomenon may not be the sole risk factor for gastric cancer. In contrast, with mimicking gastrointestinal manifestations, gastric cancer patients might have been misdiagnosed with COVID-19 in the initial stage of cancer [72]. Indeed, similar risk factors of COVID-19 and gastric cancer, such as toxins and behavioral hazards, should not be overlooked [73]. *H. pylori* has been implicated in disease pathogenesis by enhancing ACE-2 receptor expression in the gastrointestinal tract, correlating with infection duration and severity, and promoting immune dysregulation through virulent factors [74]. It is also pivotal to notice the interactive role of COVID-19 on *H. pylori* infection. Patients with H. *pylori* infection are likely more susceptible to severe COVID-19 [75], while inappropriate overuse of antibiotics such as clarithromycin and levofloxacin for COVID-19 pneumonia inclined *H. pylori* resistance in recent years [76]. Following the COVID-19 pandemic, endoscopic procedures decreased by more than 50% in the US, which can lead to a lower diagnosis of *H. pylori* infection or gastric cancer [77]. However, there is little evidence regarding the epidemiology of *H. pylori* infection in the US following the COVID-19 pandemic. Nevertheless, it appears that because of the lower conduction of endoscopic procedures and diagnostic tests for *H. pylori*, the reported numbers have decreased. Given the decline in diagnostic procedures, it is plausible that the reported numbers of *H. pylori* infections and gastric cancer cases have been underestimated. Therefore, further research is needed to assess the prevalence and impact of gastric cancer and *H. pylori* infection in the post-pandemic era. Updated studies should focus on understanding the long-term consequences of reduced diagnostic activities and the potential resurgence of gastric cancer cases as screening and healthcare services return to pre-pandemic levels.

To our knowledge, this study is one of the first studies that reported the incidence trends of gastric cancer in the US after the COVID-19 pandemic using the SEER program. However, there are some limitations. Tumor sites, including proximal, distant, cardia, non-cardia, and classification as intestinal and diffuse type, were unavailable, so they were not reported in the present study. Risk factors such as *H. pylori* infection, tobacco and alcohol use, obesity, and lack of sufficient physical activity, which are of great importance as leading factors, were also not provided. Moreover, there was a substantial impact on the health system caused by the COVID-19 pandemic, leading to decreases in cancer screening and diagnosis. This reduced the incidence rates of most cancers. It should be noted that the federal Public Health Emergency for COVID-19 in the US was expired on May 11, 2023, so the observed decline in 2020 may be influenced by factors related to the pandemic, such as delays in diagnosis and treatment, rather than a true decrease in incidence. Future studies with more extended post-pandemic data will be necessary to accurately assess the long-term effects of COVID-19 on gastric cancer trends. In this regard, the incidence data in 2020 can introduce bias into these estimates of incidence; hence, it was excluded from Joinpoint trends and only provided in graphs. We only reported sex, age, and racial/ethnic patterns of gastric cancer in the US, while other demographic variables like education, occupation, location, and insurance status were not included.

## Conclusions

Our study found that the incidence of gastric cancer varied significantly among different ages, races, and sexes. There was an increasing trend in the incidence of this cancer by age in all ethnic groups. Despite the general decrease in gastric cancer rates, there has been a rise in cases among younger adults, particularly among young Hispanic women. Additionally, a marked decline in incidence rates was recorded in 2020, coinciding with the COVID-19 pandemic. Health policymakers may use our findings to evaluate and update guidelines for gastric cancer detection in the US. Moreover, our study serves as a scaffold for future studies to determine the relevant risk factors for the observed incidence trend of gastric cancer.

## Supporting information

**S1 Table. Results of the tests of parallelism for gastric cancer incidence rate over 2000–2019 in the United States.**
(DOCX)

**S2 Table. Identical trends of gastric cancer incidence rate over 2000–2019 in the United States.**
(DOCX)

**S1 Fig. Delayed age-adjusted incidence rate of gastric cancer per 100,000 people over 2000–2019 and in 2020 in the United States, by sex.** APC: annual percent change. * Represent p-value less than 0.05.
(DOCX)

**S2 Fig. Delayed age-adjusted incidence rate of gastric cancer over 2000–2019 and in 2020 in the United States, by race/ethnicity.** APC: annual percent change. * Represent p-value less than 0.05.
(DOCX)

**S3 Fig. Delayed age-adjusted incidence rate of gastric cancer over 2000–2019 and in 2020 in the United States, by age.** APC: annual percent change. * Represent p-value less than 0.05.
(DOCX)

**S4 Fig. Delay-adjusted incidence rate of gastric cancer in the United States among males and females in each age group.** Shaded areas are the confidence interval range for the point estimates. A: all race/ethnicity; B: Hispanics; C: Non-Hispanic Blacks; D: Non-Hispanic Whites. Note: Estimates were only provided for those with more than 16 cases.
(DOCX)

**S5 Fig. Delayed age-adjusted incidence rate of gastric adenocarcinoma per 100,000 people over 2000–2019 and in 2020 in the United States, by sex.** APC: annual percent change. * Represent p-value less than 0.05.
(DOCX)

**S6 Fig. Delayed age-adjusted incidence rate of gastric adenocarcinoma over 2000–2019 and in 2020 in the United States, by race/ethnicity.** APC: annual percent change. * Represent p-value less than 0.05.
(DOCX)

**S7 Fig. Delayed age-adjusted incidence rate of gastric adenocarcinoma over 2000–2019 and in 2020 in the United States, by age.** APC: annual percent change. * Represent p-value

less than 0.05.
(DOCX)

**S8 Fig. Delayed age-adjusted incidence rate of signet ring carcinoma per 100,000 people over 2000–2019 and in 2020 in the United States, by sex.** APC: annual percent change. * Represent p-value less than 0.05.
(DOCX)

**S9 Fig. Delayed age-adjusted incidence rate of signet ring carcinoma over 2000–2019 and in 2020 in the United States, by race/ethnicity.** APC: annual percent change. * Represent p-value less than 0.05.
(DOCX)

**S10 Fig. Delayed age-adjusted incidence rate of signet ring carcinoma over 2000–2019 and in 2020 in the United States, by age.** APC: annual percent change. * Represent p-value less than 0.05.
(DOCX)

**S11 Fig. Delayed age-adjusted incidence rate of carcinoid tumor per 100,000 people over 2000–2019 and in 2020 in the United States, by sex.** APC: annual percent change. * Represent p-value less than 0.05.
(DOCX)

**S12 Fig. Delayed age-adjusted incidence rate of carcinoid tumor over 2000–2019 and in 2020 in the United States, by race/ethnicity.** APC: annual percent change. * Represent p-value less than 0.05.
(DOCX)

**S13 Fig. Delayed age-adjusted incidence rate of carcinoid tumor over 2000–2019 and in 2020 in the United States, by age.** APC: annual percent change. * Represent p-value less than 0.05.
(DOCX)

**S14 Fig. Delayed age-adjusted incidence rate of gastrointestinal stromal tumor per 100,000 people over 2000–2019 and in 2020 in the United States, by sex.** APC: annual percent change. * Represent p-value less than 0.05.
(DOCX)

**S15 Fig. Delayed age-adjusted incidence rate of gastrointestinal stromal tumor over 2000–2019 and in 2020 in the United States, by race/ethnicity.** APC: annual percent change. * Represent p-value less than 0.05.
(DOCX)

**S16 Fig. Delayed age-adjusted incidence rate of gastrointestinal stromal tumor over 2000–2019 and in 2020 in the United States, by age.** APC: annual percent change. * Represent p-value less than 0.05.
(DOCX)

## Acknowledgments

We thank the National Cancer Institute staff and its collaborators of the Surveillance, Epidemiology, and End Results Program (SEER) who prepared these data.

## Author Contributions

**Conceptualization:** Seyed Ehsan Mousavi, Seyed Aria Nejadghaderi.

**Data curation:** Armin Aslani, Seyed Ehsan Mousavi.

**Formal analysis:** Seyed Ehsan Mousavi.

**Methodology:** Armin Aslani, Amirali Soheili.

**Project administration:** Seyed Aria Nejadghaderi.

**Resources:** Seyed Ehsan Mousavi.

**Software:** Seyed Ehsan Mousavi.

**Supervision:** Seyed Aria Nejadghaderi.

**Visualization:** Seyed Ehsan Mousavi.

**Writing – original draft:** Armin Aslani, Amirali Soheili, Seyed Ehsan Mousavi, Ali Ebrahimi, Ryan Michael Antar, Zahra Yekta, Seyed Aria Nejadghaderi.

**Writing – review & editing:** Armin Aslani, Amirali Soheili, Seyed Ehsan Mousavi, Ali Ebrahimi, Ryan Michael Antar, Zahra Yekta, Seyed Aria Nejadghaderi.

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
