## [Decision Letter · Decision Letter 0]

17 Jul 2024

PONE-D-24-21194Incidence trends of gastric cancer in the United States over 2000-2020: A population-based analysisPLOS ONE

Dear Dr. Nejadghaderi,

Thank you for submitting your manuscript to PLOS ONE. After careful consideration, we feel that it has merit but does not fully meet PLOS ONE’s publication criteria as it currently stands. Therefore, we invite you to submit a revised version of the manuscript that addresses the points raised during the review process.

We look forward to receiving your revised manuscript.

Kind regards,

Deepak Dhamnetiya, MD

Academic Editor

PLOS ONE

Reviewers' comments:

Reviewer's Responses to Questions

**Comments to the Author**

1. Is the manuscript technically sound, and do the data support the conclusions?

Reviewer #1: Yes

Reviewer #2: Partly

Reviewer #3: Yes

2. Has the statistical analysis been performed appropriately and rigorously? 

Reviewer #1: I Don't Know

Reviewer #2: Yes

Reviewer #3: Yes

3. Have the authors made all data underlying the findings in their manuscript fully available?

Reviewer #1: Yes

Reviewer #2: Yes

Reviewer #3: Yes

4. Is the manuscript presented in an intelligible fashion and written in standard English?

Reviewer #1: Yes

Reviewer #2: Yes

Reviewer #3: Yes

5. Review Comments to the Author

Reviewer #1: Major Revisions

01. The federal Public Health Emergency for COVID-19 in the US was expired on May 11, 2023. As the study uses data till 2020, the effects of COVID-19 pandemic can’t be correctly evaluated.

02. Please correct reference number 4 and add the date of access as on 24th June, 2024 the rate is 36.4%

03. In page 5 the authors state: “Moreover, past research has not assessed the impact of the COVID-19 pandemic on the occurrence of gastric cancer (16, 17).”

The studies referred to were both conducted before 2020. How can the assess the impact of COVID-19 pandemic? Please rewrite this sentence.

04. Page 09: “Adenocarcinoma was the most common subtype (72.38%) in this age group.”

I could not find it in Table 1. Please add “not shown in the table” if it is not mentioned.

05. Same page: “Also, most patients were males (60.94%) and NHWs (65.82%).”

The male and female percentage does not add up to 100% and the male % is 49.86%. Please restructure table 01 to make more understandable.

06. Page 10: “Most of the cases were adenocarcinoma (74.41%)”.

I could not find it in Table 1. Please add “not shown in the table” if it is not mentioned.

07. Page 18: “Furthermore, COVID-19 led to a significant decrease in the ASIRs of gastric cancer in the US.”

Is it due to COVID-19 Pandemic or due to reporting issues? Please clarify.

08. In page 19 the authors mentioned “Despite the lower incidence of gastric cancer in the US compared to the global values, its five-year survival rate is less than 30%” which is in contrast with the earlier statement “…the survival rates for gastric cancer

remain suboptimal, with a five-year survival rate of 35.7% in the US” in page 04. Please correct this.

09. Author contribution:

Please revise as AS, AE, ZY, RMA may not fulfill the ICJME criteria.

Minor Revisions

01. Continuous line numbers should have been used in the manuscript as recommended by the journal

02. Please rephrase the statement for better understanding: “Histologically, they are classified into adenocarcinoma (accounts for approximately 90% of gastric cancers), which itself is comprised of tubular, papillary, mucinous, and poorly cohesive subtypes, in addition to other relatively rare histologic variants like gastrointestinal stromal tumors (GISTs), lymphoma, and carcinoid tumors.”

03. Page 10: “Many cases occurred in men (57.80%) and NHWs (47.04%).”

Please rewrite this line and avoid words like “many”.

04. Page 20: “In addition to the advances,

detection and treatment of H. pylori and endoscopic screening for the native and especially the

minority populations in the US are highly recommended (52).”

As the study did not consider the native and minority populations separately, can such statement be made? I recommend removing this line.

05. Page 23: “Furthermore, severe acute

respiratory syndrome coronavirus two…”

the "two" should be 2.

Reviewer #2: Please eleborate on the methods and discussion section. Also, please give a rationale, as to why this secondary analysis was done and what addition to the existing literature does this analysis bring. Use logical models and do not overarch the interpretation in the discussion.

Reviewer #3: The articles is written in a scientifically sound fashion, the title corresponds with methodology used to achieve the objective of the study. There is tittle Ethical issue as the data was generated using publicly available secondary data following sponsors' guideline. The results are correct statistically and were discussed in an intelligent manner.

The article should be accepted for publication.

6. PLOS authors have the option to publish the peer review history of their article (what does this mean?). If published, this will include your full peer review and any attached files.

Reviewer #1: No

Reviewer #2: No

Reviewer #3: **Yes: **Abdulrahman Ahmad

---

## [Author Response · Author response to Decision Letter 0]

1 Aug 2024

Response to Reviewers: 

Manuscript reference number: PONE-D-24-21194

Title: Incidence trends of gastric cancer in the United States over 2000-2020: A population-based analysis

Dear Dr. Dhamnetiya,

Thank you for giving us the opportunity to revise our paper for potential publication in PLOS ONE. We are grateful to the reviewers and the Editors for their constructive and positive comments on our paper. We have addressed each of the comments in our revision, which has further improved the article. Below we detail our changes and respond to each of the comments in turn. 

Journal requirements

Response: Thank you. We checked the guidelines and revised the manuscript accordingly. 

Response: The data availability statement was amended accordingly. 

Response: We only provided the ethics statement in the Methods.

Response: The captions for Supporting Information files were added in the manuscript.

Reviewer Comments to Author: 

Reviewer: 1

Comments to Author 

Major Revisions

01. The federal Public Health Emergency for COVID-19 in the US was expired on May 11, 2023. As the study uses data till 2020, the effects of COVID-19 pandemic can’t be correctly evaluated.

Response: Thank you so much for your time and your valuable feedback. We acknowledge that our data collection concludes in 2020, which limits our ability to fully evaluate the long-term effects of the COVID-19 pandemic on gastric cancer incidence. Our objective was to provide preliminary observations on the immediate impact of the pandemic within the available timeframe. We have noted this limitation in our discussion and clarified that the observed decline in 2020 may be influenced by factors related to the pandemic, such as delays in diagnosis and treatment, rather than a true decrease in incidence. Future studies with more extended post-pandemic data will be necessary to accurately assess the long-term effects of COVID-19 on gastric cancer trends. We provided the explanations in the last paragraph of Discussion. 

02. Please correct reference number 4 and add the date of access as on 24th June, 2024 the rate is 36.4%

Response: It was updated accordingly. 

03. In page 5 the authors state: “Moreover, past research has not assessed the impact of the COVID-19 pandemic on the occurrence of gastric cancer (16, 17).”

The studies referred to were both conducted before 2020. How can the assess the impact of COVID-19 pandemic? Please rewrite this sentence.

Response: We want to highlight that no similar studies used the recent data to evaluate the effects of COVID-19 pandemic on incidence trends of gastric cancer by age, sex, and race. The cited references are those recent relevant articles on this topic that did not evaluate the COVID-19 pandemic. We rewrote the sentence for clarification. 

04. Page 09: “Adenocarcinoma was the most common subtype (72.38%) in this age group.”

I could not find it in Table 1. Please add “not shown in the table” if it is not mentioned.

Response: The ratios provided in the tables are all calculated using the race-specific population as the denominator, regardless of the age range. However, it's important to note that in the manuscript, for any age-specific ratio, both race and age-specific populations were used as the denominator, as accurately mentioned. As it was not shown in the table, we clarified it in the text. 

05. Same page: “Also, most patients were males (60.94%) and NHWs (65.82%).”

The male and female percentage does not add up to 100% and the male % is 49.86%. Please restructure table 01 to make more understandable.

Response: The gender ratios in the table were calculated by dividing the age-specific sex values by the total population of the corresponding sex, however in the manuscript in order to calculate the age-specific sex ratio, only age-specific population was used. This is the reason for the confusion.

We used the following formula in the table for each entry:

We used the following formula for age-specific entries in the manuscript:

In similar fashion, in the manuscript in order to calculate the age specific racial ratio only age specific population was used.

06. Page 10: “Most of the cases were adenocarcinoma (74.41%)”.

I could not find it in Table 1. Please add “not shown in the table” if it is not mentioned.

Response: We provided further details about its calculation in the manuscript in the previous comment and it was clarified that it was not shown in the table, in the main text. 

07. Page 18: “Furthermore, COVID-19 led to a significant decrease in the ASIRs of gastric cancer in the US.”

Is it due to COVID-19 Pandemic or due to reporting issues? Please clarify.

Response: It may be attributed to multiple factors related to the COVID-19 pandemic. These factors include disruptions in healthcare services, such as reduced access to routine screenings and diagnostic procedures, as well as delays in seeking medical attention due to lockdowns and fear of contracting the virus. Additionally, reporting issues and temporary reallocation of healthcare resources towards managing the pandemic may have contributed to the apparent decline in reported cases. The explanations were provided in the first paragraph of Discussion. 

08. In page 19 the authors mentioned “Despite the lower incidence of gastric cancer in the US compared to the global values, its five-year survival rate is less than 30%” which is in contrast with the earlier statement “…the survival rates for gastric cancer

remain suboptimal, with a five-year survival rate of 35.7% in the US” in page 04. Please correct this.

Response: Thank you. It was corrected. 

09. Author contribution:

Please revise as AS, AE, ZY, RMA may not fulfill the ICJME criteria.

Response: Per journal guidelines (https://journals.plos.org/plosone/s/file?id=wjVg/PLOSOne_formatting_sample_main_body.pdf), we removed this section and just provided the contribution of each author in the submission site. We considered this point in the revision.

Minor Revisions

01. Continuous line numbers should have been used in the manuscript as recommended by the journal

Response: It was added.

02. Please rephrase the statement for better understanding: “Histologically, they are classified into adenocarcinoma (accounts for approximately 90% of gastric cancers), which itself is comprised of tubular, papillary, mucinous, and poorly cohesive subtypes, in addition to other relatively rare histologic variants like gastrointestinal stromal tumors (GISTs), lymphoma, and carcinoid tumors.”

Response: To improve clarity, we will rephrase the statement as follows:

"Histologically, gastric cancers are primarily classified into adenocarcinoma, which accounts for approximately 90% of cases. Adenocarcinoma itself has several subtypes, including tubular, papillary, mucinous, and poorly cohesive types. In addition to adenocarcinoma, there are other less common histologic variants such as gastrointestinal stromal tumors (GISTs), lymphomas, and carcinoid tumors".

03. Page 10: “Many cases occurred in men (57.80%) and NHWs (47.04%).”

Please rewrite this line and avoid words like “many”.

Response: We revised the sentence to:

"A total of 57.80% of cases occurred in men and 47.04% occurred in NHWs".

04. Page 20: “In addition to the advances,

detection and treatment of H. pylori and endoscopic screening for the native and especially the

minority populations in the US are highly recommended (52).”

As the study did not consider the native and minority populations separately, can such statement be made? I recommend removing this line.

Response: As recommended, we removed the sentence. 

05. Page 23: “Furthermore, severe acute

respiratory syndrome coronavirus two…”

the "two" should be 2.

Response: It was corrected. 

Reviewer: 2

Comments to Author 

Please eleborate on the methods and discussion section. Also, please give a rationale, as to why this secondary analysis was done and what addition to the existing literature does this analysis bring. Use logical models and do not overarch the interpretation in the discussion.

Response: Thank you so much for your time and your valuable comments. We provided the rationale for this study and how the findings might be helpful for health policymakers in the last paragraph of Introduction. We also expanded Methods and Discussion by providing further details about the methodology and statistical tests performed in the study. 

Reviewer: 3

Comments to Author 

The articles is written in a scientifically sound fashion, the title corresponds with methodology used to achieve the objective of the study. There is tittle Ethical issue as the data was generated using publicly available secondary data following sponsors' guideline. The results are correct statistically and were discussed in an intelligent manner.

The article should be accepted for publication.

Response: Thank you so much for your kind words and positive feedback on our manuscript. As it was mentioned in the Methods section, the access to the SEER data was in accordance with the SEER data agreement and after approval of the relevant guidelines.

---

## [Decision Letter · Decision Letter 1]

22 Aug 2024

Incidence trends of gastric cancer in the United States over 2000-2020: A population-based analysis

PONE-D-24-21194R1

Dear Dr. Nejadghaderi,

We’re pleased to inform you that your manuscript has been judged scientifically suitable for publication and will be formally accepted for publication once it meets all outstanding technical requirements.

Kind regards,

Deepak Dhamnetiya, MD

Academic Editor

PLOS ONE

Reviewers' comments:

Reviewer's Responses to Questions

**Comments to the Author**

1. If the authors have adequately addressed your comments raised in a previous round of review and you feel that this manuscript is now acceptable for publication, you may indicate that here to bypass the “Comments to the Author” section, enter your conflict of interest statement in the “Confidential to Editor” section, and submit your "Accept" recommendation.

Reviewer #1: All comments have been addressed

Reviewer #2: All comments have been addressed

2. Is the manuscript technically sound, and do the data support the conclusions?

Reviewer #1: Yes

Reviewer #2: Yes

3. Has the statistical analysis been performed appropriately and rigorously? 

Reviewer #1: Yes

Reviewer #2: Yes

4. Have the authors made all data underlying the findings in their manuscript fully available?

Reviewer #1: Yes

Reviewer #2: Yes

5. Is the manuscript presented in an intelligible fashion and written in standard English?

Reviewer #1: Yes

Reviewer #2: Yes

6. Review Comments to the Author

Reviewer #1: Thank you for addressing all my comments. The manuscript can be published now. I recommend publication.

Reviewer #2: Thank you for making the requested changes by the reviewers. The manuscript is in better shape now and can be published.

7. PLOS authors have the option to publish the peer review history of their article (what does this mean?). If published, this will include your full peer review and any attached files.

Reviewer #1: No

Reviewer #2: **Yes: **Aftab Ahmad

---

## [Editor Report · Acceptance letter]

15 Sep 2024

PONE-D-24-21194R1 

PLOS ONE

Dear Dr. Nejadghaderi, 

I'm pleased to inform you that your manuscript has been deemed suitable for publication in PLOS ONE. Congratulations! Your manuscript is now being handed over to our production team.

Kind regards, 

on behalf of

Dr. Deepak Dhamnetiya 

Academic Editor

PLOS ONE